# Optimal Adaptive Electrode Selection to Maximize Simultaneously Recorded Neuron Yield

**John Choi**
Center for Neural Science
New York University
jc4007@nyu.edu

**Krishan Kumar**
Center for Neural Science
New York University
ksk434@nyu.edu

**Mohammad Khazali**
Center for Neural Science
New York University
m.khazali@nyu.edu

**Katie Wingel**
Center for Neural Science
New York University
kew445@nyu.edu

**Mahdi Choudhury**
Center for Neural Science
New York University
mc5368@nyu.edu

**Adam Charles**
Department of Biomedical Engineering
Johns Hopkins University
adamsc@jhu.edu

**Bijan Pesaran**
Center for Neural Science
New York University
bijan@nyu.edu

## Abstract

Neural-Matrix style, high-density electrode arrays for brain-machine interfaces (BMIs) and neuroscientific research require the use of multiplexing: Each recording channel can be routed to one of several electrode sites on the array. This capability allows the user to flexibly distribute recording channels to the locations where the most desirable neural signals can be resolved. For example, in the Neuropixel probe, 960 electrodes can be addressed by 384 recording channels. However, currently no adaptive methods exist to use recorded neural data to optimize/customize the electrode selections per recording context. Here, we present an algorithm called classification-based selection (CBS) that optimizes the joint electrode selections for all recording channels so as to maximize isolation quality of detected neurons. We show, in experiments using Neuropixels in non-human primates, that this algorithm yields a similar number of isolated neurons as would be obtained if all electrodes were recorded simultaneously. Neuron counts were 41-85% improved over previously published electrode selection strategies. The neurons isolated from electrodes selected by CBS were a 73% match, by spike timing, to the complete set of recordable neurons around the probe. The electrodes selected by CBS exhibited higher average per-recording-channel signal-to-noise ratio. CBS, and selection optimization in general, could play an important role in development of neurotechnologies for BMI, as signal bandwidth becomes an increasingly limiting factor. Code and experimental data have been made available[1].

## 1 Introduction

Modern neurotechnologies operate across a range of physical modalities (such as light, electricity, magnetism) and offer the potential to record the activity of thousands or even millions of neurons [7]. High performing brain-machine interfaces (BMIs), a type of neurotechnology, require simultaneous

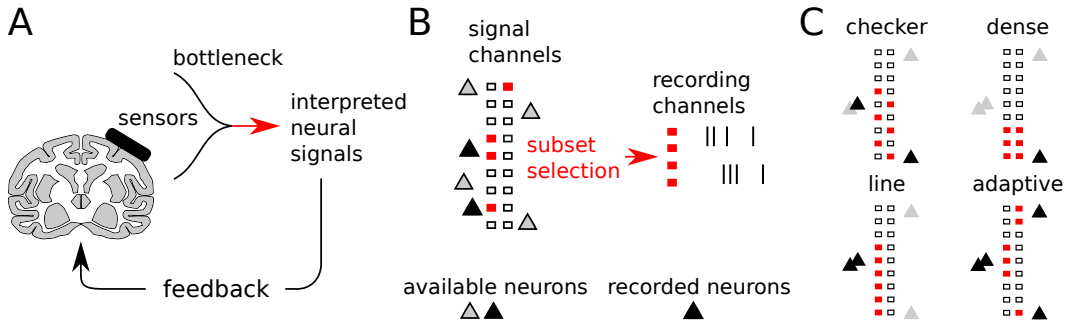

Figure 1: Signal channel selection problem. A: Brain-machine interfaces must record from large numbers of signal channels simultaneously, and therefore face a bottleneck. B: "Recording channels" are sub-sampled from a larger set of "signal channels." C: Adaptive selection maximizes the number of recorded neurons, while naïve selection strategies fail in this regard.

recording and processing of large populations of neurons, and therefore suffer a bandwidth bottleneck: Engineering requirements constrain the number of signal channels that can be acquired simultaneously (Fig. 1A).

Bandwidth bottlenecks lead to an inherent problem of how to select from the available signal channels in order to optimally monitor the neural population of interest [20] (Fig. 1B). For example, selecting maximally informative populations for muscle activation can improve BMI system performance [6, 21], which requires maximizing the size of the recorded neuronal ensemble. Such selection is constrained by the nuances of engineering solutions, such as power for wireless arrays [27], optical paths and dwell times for microscopes [22] and wiring constraints for electrode arrays [23, 17, 1].

If there were no constraints, one could simply measure from all available signal channels. There would be no selection problem. If there were too many constraints, there would be very few simultaneously recordable signal channels. For most modern neurotechnologies, however, the space of possible selections is combinatorial. For example, Neuropixel probes [5] contain 960 electrodes (signal channels). However, only 384 recording channels can be acquired simultaneously. Subject to other constraints (see below), for this array, there are $2.5^{149}$ different possible selections.

In the face of such combinatorial explosion, heuristics have prevailed. For example, previously published heuristics for the Neuropixel array either route all recording channels to groups of contiguous electrodes in order to densely sample from a small segment of the probe or skip every other electrode to form a checkerboard or a sparse linear pattern (Fig. 1C).

These heuristics are limited for several reasons. Neither selects electrodes to maximize neuronal ensemble size based on where detected neurons appear. The density of sampling is uniform for dense and checkerboard patterns, leading to surplus density in some regions and lack of density in others. Also, the selection does not consider signal quality or waveform variability. If one electrode is noisier than similar neighbors, then it should not be recorded.

Optimal selection of recording sites is a general problem when designing a sensing system with costly engineering constraints, such as RADAR arrays [15], seismic wavefield sensing [13] and magnetic resonance imaging [11]. Many applications reduce to determining optimal sensor placements for any signal from a given class, e.g., randomized compressive sampling for sparse data [3]. Recording channel selection instead must be optimized for a set of signal sources, and so is more related to subspace identification (e.g., PCA) and optimal design [16]. Specifically, the subspace of signal channels that are best captured by the limited number of recording channels must be determined. The unique constraints in Neural Matrix-style recording devices [1], which implement subset selection through multiplexing, require more specialized algorithms. Moreover, the typical cost functions used in subspace estimation and optimal design fail to capture the information of importance to neural recordings: isolating many single neurons. It is therefore necessary to create a new class of algorithms that can meet the needs of optimal adaptive sub-sampling in next-generation neural recording devices.

With the philosophy of optimal channel selection in mind, we demonstrate efficient, automated procedures for electrode selection applied to a state-of-the-art commercially available probe, the

Neuropixel (Imec). Using these methods, we show channel selections that are able to simultaneously resolve most spiking neurons present around the probe in recordings of macaque premotor cortex.

## 2 Adaptive electrode selection

Modern electrode arrays [5, 1] implement subset selection in the form of multiplexing, as depicted in Figure 2A. Under this scheme, a recording channel can be assigned to one of several electrode sites on the probe, corresponding to "banks" of spatially adjacent electrodes. For example, on the Neuropixel Phase 1 probe, each of 384 channels can be assigned to one of 2-3 banks spanning 960 electrodes, with the number of possible selections given by $(3^{192})(2^{192}) \sim 2.5 \times 10^{149}$. In this report, we discuss the problem of optimally assigning channels to banks so as to jointly maximize the number of neurons resolved by the array. We present a data-driven method and describe several benchmark heuristics.

We first denote a multichannel neuronal spike waveform as the sequence of random vectors $v(t) \in \mathbb{R}^E$ for $t = 1, ..., T$, where $T$ is the number of time points in the waveform, and $E$ is the total number of available electrodes. We can never observe all elements of $v(t)$ simultaneously, and instead we can only observe $N_c < E$ recording channels at a time. Since these channels require hardware multiplexing, the available electrodes are subdivided into $B$ banks. Hence each channel $c$ can be addressed to one of $B$ distinct sites. We represent the address state of every channel with a data structure $\theta$ we refer to as the *selection map*, which contains $N_c$ assignments of the form $\theta_{c \to b}$, where $b \in \{1, \cdots, B\}$. For example, in Figure 2A, a channel is shown addressed to the lowermost bank, thereby "enabling" the corresponding electrode.

If an electrode is not addressed to, i.e., it is "disabled," then the corresponding entries of $v$ are removed, and we denote the remaining vector by $v^\theta$. We then stack voltages from all time points to form a feature vector $x^\theta = [v^\theta(1)^T, ..., v^\theta(T)^T]^T$. Examples of $x^\theta$ serve as input to an automated spike sorter. For a given recording using $\theta$ as a selection map, the spike sorter returns the spike times and neuron identities for $N_\theta$ discriminated neurons.

The ultimate goal is to find a $\theta$ that maximizes $N_\theta$. This objective is time-consuming to evaluate for each candidate selection map $\theta$, since one evaluation requires gathering a recording using $\theta$ and sorting the spikes from the recording. Therefore, we seek a proxy that allows us to optimize the selection map efficiently.

Herein, we describe a method (Fig. 2B) that initially takes $B$ dense pilot recordings of relatively short duration, where each recording samples an entire bank at full resolution. That is, for the $b$th recording, for all channels $c$ addressable to bank $b$, the selection map contains $\theta_{c \to b}$. These pilot recordings are then spike sorted, and the spike waveforms from $N_s$ randomly chosen spikes from each isolated neuron serve as input to our selection method.

### 2.1 Classification-based selection (CBS)

A selection algorithm chooses which electrodes on the array to enable and disable. This choice affects the ultimate ability to detect and discriminate waveforms from neurons present around the array. Therefore, we seek to choose the selection based on how well the enabled electrodes differentiate spikes from one neuron from spikes from all other neurons. This would maximize neuron separability but would also prioritize channels with higher signal-to-noise, since they would be more reliable for discrimination.

We therefore approximate the optimization of $N_\theta$ with an optimization of discriminability across all available neurons. Here, we estimate the total number of available neurons with those extracted from dense recordings of each bank, thereby yielding $N$ available neurons (Fig. 2B). We denote $J(.)$ as a metric of discriminability across these $N$ neurons. The problem then becomes:

$$\max_\theta J(\theta).$$

In other words, we pose the problem of electrode selection as constrained feature selection. The "features" are the enabled electrodes. We can train a classifier to predict neuron labels from $x^\theta$ using a candidate selection and use the classifier's objective as a measure of the selection's quality. We can then perform a greedy search over each channel's bank assignments. This, however, involves evaluating the classifier's objective repeatedly, perhaps requiring thousands of evaluations. Therefore,

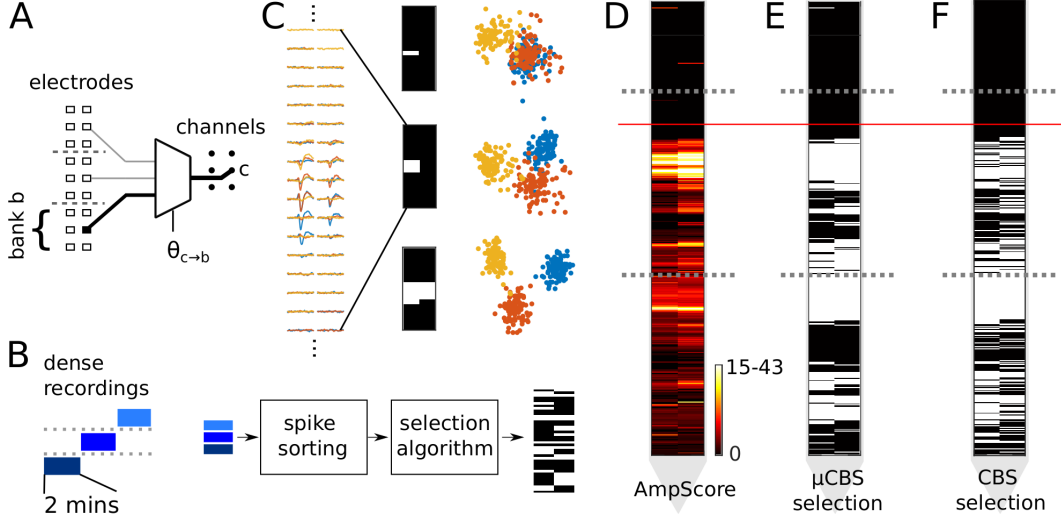

Figure 2: Electrode selection process. A: Diagram of spatial multiplexing, a type of subset selection found in modern electrode arrays. B: Experimental procedure for using dense single bank recordings as input to a selection algorithm. C: Illustration of waveform separability as a function of electrode selection. Three exemplar neurons' spike waveforms in a small segment of an array can be more separable depending on the electrodes that are selected. (left panel) Mean spike waveforms. (right panel) Three different selections on this array segment (white pixels indicate enabled electrodes) and low-dimensional projections of the spike waveforms. D: Example relative amplitude scores accumulated for every electrode. Dotted lines indicate bank boundaries, and the red line shows the estimated pial surface level. E-F: The corresponding enabled (white pixels) electrode selections for the $\mu$CBS and CBS methods.

although any separability criterion can be used, desirable criteria are ones that can be evaluated quickly for different subsets of electrodes.

$x^\theta$, for a desirable selection map, will exhibit low variability across spikes from the same neuron and high variability across spikes from different neurons. Since spike waveforms are often Gaussian given the neuron label, with known exceptions due to bursting, drift, and overlapping spikes, class discriminability can be completely summarized by using the within-class and between-class covariances, and therefore, the metric of discriminability $J$ should be a function of their ratio.

More precisely, denoting spike waveforms from neuron $i$ as $x_i^\theta$, we define the within- and between-neuron "scatter" matrices as

$$S_w^\theta = \frac{1}{N} \sum_{i=1}^{N} \mathbf{E}\left[(x_i^\theta - \mu_i^\theta)(x_i^\theta - \mu_i^\theta)^T\right] = \frac{1}{N} \sum_{i=1}^{N} \Sigma_i^\theta,$$

$$S_b^\theta = \frac{1}{N} \sum_{i=1}^{N} (\mu_i^\theta - \mu^\theta)(\mu_i^\theta - \mu^\theta)^T,$$

where $\mu_i^\theta$, $\Sigma_i^\theta$ are the within-class sample mean and covariance of $x_i^\theta$, and $\mu^\theta$ is the sample mean waveform across all neurons. We then define the objective as

$$J(\theta) = \mathrm{Tr}\left[(S_w^\theta)^{-1} S_b^\theta\right]. \tag{1}$$

This objective can be interpreted as a signal-to-noise ratio (SNR), where the between-class covariance is the "signal" and the within-class covariance is the "noise." This objective is the trace criterion commonly used in Linear Discriminant Analysis (LDA) [4]. For classification or visualization, we can define $z^\theta = U^T x^\theta$ as a projection of $x^\theta$ into a subspace that maximizes this SNR. The matrix $U \in \mathbb{R}^{n \times m}, m = N - 1$ is formed from the $m$ leading eigenvectors of $(S_w^\theta)^{-1} S_b^\theta$ arranged horizontally as columns. See Figure 2C, where examples of $z^\theta$ are plotted for three neurons under different selections.

This objective needs to be evaluated for every candidate selection map, $\theta$. Importantly, evaluating a new selection does not require recomputing the scatter matrices. If we were able to form scatter matrices with all electrodes enabled, denoted without superscripts as $S_w$ and $S_b$, then evaluating $J(\theta)$ amounts to simply removing the rows and columns of $S_w$ and $S_b$ corresponding to disabled electrodes prior to solving the linear system in (1), which requires $O(T^3 E^3)$ flops. $S_w$ and $S_b$ of course cannot be fully computed due to the problem's constraints. However, we can approximate these matrices with the dense pilot recordings: If we assume for the neurons identified in the $b$th recording, that unrecorded portions of $v(t)$ are 0, then $S_w, S_b$ are block diagonal along bank boundaries. Therefore, each bank can be treated independently and $J(\theta)$ can be evaluated for each bank separately and accumulated, i.e., $J(\theta) = J^1(\theta) + ... + J^B(\theta)$, where $J^b(.)$ is the objective evaluated for subsets of elements of $x, S_w$ and $S_b$ as well as for subsets of neurons that correspond with bank $b$. This simplification reduces the run time to $O(BT^3 N_c^3)$ per iteration. Since these operate on pre-formed scatter matrices, the run time does not depend on the number of spikes.

To optimize $J(\theta)$, we perform a random greedy swap search in the space of selection maps. First, we initialize $\theta$ randomly or with one of the preset selection maps (e.g. a checkerboard map). Then, in a manner similar to Gibbs sampling, we pick a channel $c$ at random (without replacement) and evaluate $J(.)$ for each value in $\sigma(\theta, c) = \{\theta_{c \to 1}, ..., \theta_{c \to B}\}$, where $\theta_{c \to b}$ is the selection $\theta$, but with the $c$th channel routed to bank $b$. Then, on each iteration,

$$\theta \leftarrow \text{argmax}_{\lambda \in \sigma(\theta, c)} J(\lambda). \tag{2}$$

This channel-by-channel maximization is repeated for multiple passes through all channels until $J$ ceases to change within a pass. This algorithm requires that each subproblem (2) have a unique minimum, i.e., no ties, in order to converge in $\theta$ as well as converging in $J$. However, it is unlikely, for an array that has at least $B - 1$ banks monitoring distinct sets of neurons, that this uniqueness condition is ever violated.

Despite the recording-wise parallelizations, the evaluation of (1) remains high-dimensional and computationally intensive. We thus introduce the two simplifications that greatly improve tractability for typical problem sizes.

**Simplification 1: using channel-wise PCA to reduce dimensionality** Temporal autocorrelations in the spike waveform mean that the dimensionality of $x$, $n = TE$ is unnecessarily large (for the Neuropixel Phase 1 probe using a 2 ms spike waveform window, $n = (61)(960) = 58,560$), and solving the system in (1) can be infeasible. As is commonly done in spike-sorting algorithms [2, 14, 18], principal component analysis (PCA) is first performed on each channel's waveforms across all spikes that involve this channel. We can then choose the first $r$ principal component scores for each channel, thereby reducing the dimensionality to $n = rE$.

**Simplification 2: exploiting bandedness** With large banks, the linear system in (1) can be quite large. In this case, we can exploit the spatial layout of the bank. Specifically, the effective extent of the voltage from one neuron does not spread across an entire bank. If a neuron's spread within a bank is limited, and all channels are ordered according to their spatial arrangement, then $S_w^\theta$ is sparse and banded. With both simplifications, solving the system in (1) only requires $O(kBr^2 N_c^2)$ flops, where $k$ is the bandwidth.

## 2.2 Fast approximation to CBS using diagonal covariance ($\mu$CBS)

Optimizing $J(\theta)$ in general requires multiple passes through the array, since optimizing one channel, as in (2), might alter the decision to reassign another channel. For applications that require deterministic run times, we also describe a variant of CBS called $\mu$CBS that approximately maximizes (1) assuming only diagonal covariance structures. This simplification lets us treat each channel's selection problem as independent from other channels. Therefore, repeated passes through the array are not required.

We first apply a diagonal assumption on $S_w$, i.e., $[S_w]_{ij} \to 0$ if $i \neq j$. Then, the full trace (no selection map applied yet) in (1) becomes

$$\text{Tr}\left[S_w^{-1} S_b\right] = \sum_{j=1}^n \frac{\sum_{i=1}^N (\mu_i^{(j)} - \mu^{(j)})^2}{\sum_{i=1}^N \mathbf{E}[(x_i^{(j)} - \mu_i^{(j)})^2]},$$

where $\mu$, $\mu_i$ denotes full (not subset-selected) averages of $x$, and superscript $(j)$ denotes indexing the $j$th element. For simplicity of notation, we define the ratio as $R(j) = \sum_{i=1}^{N} (\mu_i^{(j)} - \mu^{(j)})^2 / \sum_{i=1}^{N} \mathbf{E}[(x_i^{(j)} - \mu_i^{(j)})^2]$, which can be interpreted as a signal-to-noise ratio for each element $j$. For a given selection map $\theta$, we can modify $J(\theta) \rightarrow J_\mu(\theta)$ as

$$J_\mu(\theta) = \sum_{e \in \text{Enabled}(\theta)} \sum_{j \in \phi(e)} R(j), \tag{3}$$

where $\text{Enabled}(\theta)$ is the set of electrodes enabled by $\theta$, and $\phi(e) = \{e, e + E, ..., e + (T-1)E\}$ is the set of indices of $x$ that pertain to electrode $e$. Enabling (or disabling) an electrode amounts to including (or excluding) that electrode's contribution to $J_\mu(\theta)$. Importantly, the effect on $J_\mu(\theta)$ does not depend on the enabled-state of any other electrodes. Therefore, (3) can be optimized for every channel independently. This observation enables a simple algorithm. First, we assign all electrodes $e$ a score we call $\text{AmpScore}(e)$ based on the sum of $R$ values observed on that electrode:

$$\text{AmpScore}(e) = \sum_{j \in \phi(e)} R(j). \tag{4}$$

Then, for each channel, we choose the bank assignment that corresponds with an electrode with the highest available AmpScore. After a single pass, $J_\mu(\theta)$ is fully optimized. Since solving the full system in (1) is no longer required, there is less need for reducing the dimensionality of $x$ with PCA, and $\mu$CBS can operate in full waveform space.

## 2.3 Benchmark selection methods

**Single-bank**   In this method, all channels are assigned to one contiguous bank of electrodes, thereby densely sampling it (Fig. 1C). This strategy has maximum electrode density, but has poor spatial coverage. Depending on the probe design, a dense selection might oversample some areas.

**Line**   This method corresponds to selecting one contiguous column of contacts for each bank of electrodes. This sacrifices resolution in one dimension for coverage in another.

**Checker**   This method forms a checkerboard pattern of enabled electrodes. This halves the resolution evenly throughout the sampled area, but doubles the spatial coverage.

## 3   Empirical assessment

The sampling methods were tested and compared on Neuropixel (Imec, Phase 1) recordings from the premotor cortex of an awake macaque. We made dense recordings of each bank and simulated an experimental procedure where the first two minutes of data would be used as a training set for the selection algorithms, as depicted in Figure 2B. After optimization, the selection would be tested by simulating the selection on full-length bank recordings by removing channels that were not assigned to by the algorithm. Both the training and full-length recordings were spike-sorted automatically using Kilosort2 (`https://github.com/MouseLand/Kilosort2`). The complete bank recordings ranged from 2 to 17 minutes, across 10 recording sessions and 3 banks. Probes were lowered to 6-7 mm below the cortical surface, and only encountered spiking activity in the deepest two-thirds of electrodes (see Figure 2D). Analysis was restricted to the first two banks, numbered 0 and 1.

In every bank recording, for each neuron discovered by the automated spike sorter, $N_s = 100$ spike waveforms were extracted at random. Neurons with average firing rate less than 100 spikes per 2 minutes (0.8 Hz) were excluded from CBS and from the validations described below. Each spike waveform's DC offset was removed by subtracting the median voltage across the duration of the waveform (2 ms). This was followed by high-pass filtering at 150 Hz and applying a spatial whitening matrix to the voltage as in [14] to spatially sharpen the multi-channel waveforms. Herein, when we discuss the spike waveforms, we are referring to the preprocessed waveforms. For the channel-wise PCA step, we set $r \leftarrow 3$. $\mu$CBS was used in full-waveform space (no PCA) and its waveforms were conditioned with median subtraction and high-pass filtering with the same parameters as CBS.

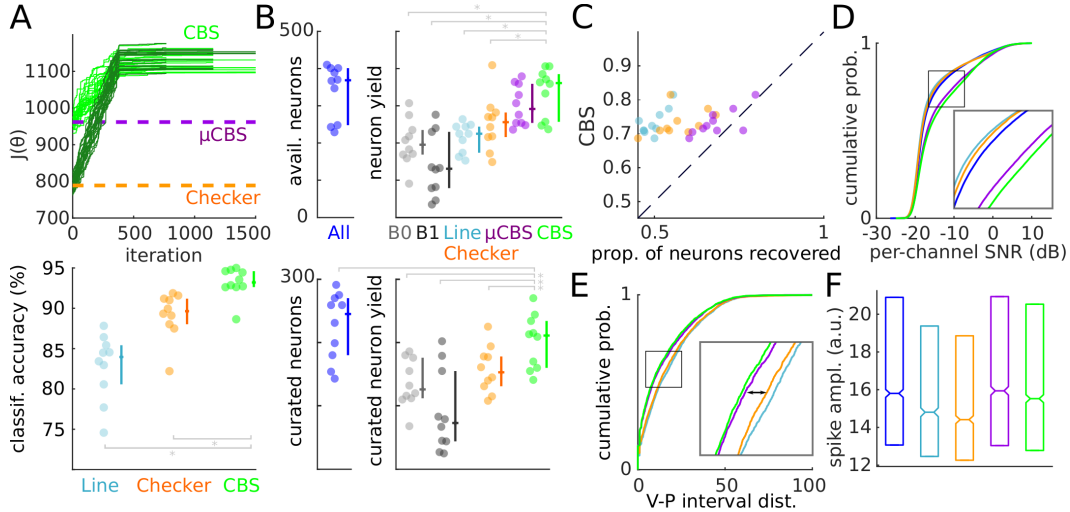

Figure 3: Selection method comparisons. A: (top panel) CBS-based optimization criterion $J(\theta)$ trace for a representative session. Different shades of green indicate different initialization methods, and each curve is a realization (see text). (bottom panel) shows the corresponding correct classification rate under cross-validation. Each point in a group represents one session. Bars show median with quartiles. Bracket: p<0.05, *: p<0.01, Wilcoxon signed rank test. B: (top panel) Number of available neurons detected on all banks (All), and the number of neurons discovered for each selection method. B0,1: dense selection of bank 0 or 1. Only significant differences from CBS and $\mu$CBS are shown. (bottom panel) Human-curated neuron yield. C: Proportion of available neurons "recovered" by each selection method. Vertical axis is the proportion recovered by CBS, and the horizontal is the recovery rate for all other selection methods. All differences from CBS were significant. D: Empirical cumulative distribution functions (CDFs) over the SNR of each channel-neuron pair observed for each selection method. All pairs of distributions were found to have different medians (p<0.0001, 2-sample KS test and rank-sum test). E: CDF for the Victor-Purpura (V-P) interval distance from each All neuron to the nearest sub-sampled neuron in spike train space. Double-sided arrow indicates statistically significant difference in distributions (p<0.001, 2-sample KS test). Unmarked differences greater than this effect were also significant. F: Neuronal spike amplitudes for each selection method. Bars indicate median with quartiles. Notches signify 95% confidence intervals of the median.

## 3.1 CBS electrode selections

Typical selections for the CBS and $\mu$CBS algorithms are shown in Figure 2E-F. Figure 2D shows AmpScores as in (4) accumulated across the electrode array. Figure 2E shows the resulting $\mu$CBS selection. The CBS algorithm's resulting selection is shown in Figure 2F.

The CBS-based algorithm's trace of the objective, $J(\theta)$, over 25 realizations are shown in Figure 3A for a representative recording. Realizations were generated from different randomly chosen sets of $N_s$ spikes per neuron and different sequence of channels chosen for (2). The terminal selection was found within 2-5 passes of the array. Notably, most of the improvement occurred in the first pass.

**Waveform separability** We also measured how well a spike waveform could be classified as belonging to a certain neuron under different selection strategies. In this assessment, the neuron classes are determined from the dense recordings, and hence waveform separability is solely a function of the strategy's selection map $\theta$. For each map, we trained a nearest-mean classifier in LDA-projected space (the space of $z^\theta$) using $N-1$ dimensions. Figure 3A (bottom panel) compares the classification accuracy (10 Monte-Carlo selections of 75% train, 25% test) resulting from different selection strategies. CBS achieved a median test accuracy of 93.1% correct, which was significantly higher than Checker's performance of 89.6% (p=0.002, signed rank test, n=10 sessions). CBS's terminal performance did not significantly depend on the initialization method (p=0.557, signed rank test). Across realizations, the standard deviation of classification accuracies was 0.25%, showing only a slight dependence on initialization.

**Neuron yield and properties**   Under automated spike-sorting, the CBS and $\mu$CBS methods yielded similar numbers of discovered neurons as would be detected if the whole array were densely recorded (a fictitious selection we refer to as "All"), while the preset heuristics discovered substantially fewer. The number of available neurons for each session, along with the neuron yields for each algorithm, is shown in Figure 3B. Each point in a group represents a recording session. The median number of discovered neurons across sessions was 368.5 for All. CBS and $\mu$CBS had a median yield 361 and 291.5 neurons, respectively. This was 41% and 14% more neurons than discovered using Checker (256 neurons), and 84.7% and 49.1% more than when sampling bank 0 densely (195.5 neurons, abbreviated B0 in Figure 3B). Line and Checker methods both discovered significantly fewer neurons than existed in the All case (p=0.002, signed rank test, FDR=0.0033 for Line, and p=0.014, FDR=0.017 for Checker, n=10 sessions), but no significant difference from All was detected for CBS or $\mu$CBS (p=0.25 for $\mu$CBS, p=1.0 for CBS, signed rank test). When taken as a percentage of All's yield for each recording, CBS's neuron yield was $99.7 \pm 10.3\%$ (mean $\pm$ std. dev.), which was significantly higher (p=0.0039, signed rank test) than Checker's yield of $74.3 \pm 14.9\%$ and Line's yield of $63.8 \pm 5.6\%$.

These trends were also present when spike sorting was followed by human curation. A labeler was instructed to use a graphical curation tool (Phy) to split/merge neuron identities, and to label a subset as "good," well-isolated single neurons. The median number of curated units was 245. CBS yielded 211, whereas Checker yielded 153 (see Figure 3B lower panel). Both methods yielded significantly less than All (p=0.011 for CBS, p=0.002 for Checker, signed rank test). When taken as a percentage of All's curated yield from each session, CBS's neuron yield was $89.9 \pm 8.5\%$, which was significantly (p=0.002, signed rank test) higher than Checker's yield of $71.0 \pm 13.7\%$.

We then examined how many All neurons could be "recovered" by the different algorithms. We define an All neuron to be recovered if its temporal spiking pattern overlaps with that of a sub-sampled neuron at least 80% of the time, up to 2 ms accuracy. An All neuron can only be recovered once. The neuron recovery rate for CBS was found to be 73.0% (median). CBS performed 8.4% better in this regard than $\mu$CBS (67.3%, p=0.013, signed rank test), 29.2% better than Checker (56.5%, p=0.002), and 47.0% better than Line (49.6%, p=0.002). This comparison is shown in Figure 3C. CBS also recovered more All neurons than B0 or B1, outperforming them by 11.7% and 111.7%, respectively. The difference with B0 was only detected to a 91.6% confidence level (signed rank test).

The neurons discovered by CBS had higher per-channel SNR than those discovered other selection methods. For every channel-neuron pair identified by the spike sorter, we define the SNR as the $\log_{10}$ of the squared norm of the waveform mean divided by the waveform variance. The median SNR across channel-neuron pairs was 1.6 dB higher for the CBS selection than for All (p<0.00001, rank-sum test). The distribution function over SNR values is shown in Figure 3D. All methods had statistically different distribution functions (p<0.00001, 2-sample KS test) from each other.

As further confirmation, we compared each method's spike times with All neuron spike times through the Victor-Purpura (V-P) interval distance [25] (related to the Earth-Mover's distance). For each All neuron, we computed the nearest sub-sampled neuron in spike-train space. We used $q = 0.75$ (cost/second factor in [25]) and a duration of analysis set to the time of the 100th (minimum spike count across recordings) spike. The distribution of minimum distances for each selection method is shown in Figure 3E. CBS's median distance was 32.8% lower than Checker's (p<0.00001, signed rank test).

If a selection method undersamples a neuron, then it is quite possible that estimates of the neuron's spike amplitudes will be lower than when the neuron is sampled densely. To test if this occurred, we measured neuron amplitude distributions for each method (Fig. 3F). We define the spike amplitude as the scaling factor that would need to be applied to a unit-norm multichannel template to recreate the spike waveform, as in [14]. Median neuron amplitudes for CBS ($\mu$CBS) were 98.2% (100.8%) that of All's median amplitude. Heuristic methods showed lower neuronal amplitudes: 91.2% for Checker and 93.7% for Line. These were both significantly lower than with CBS/$\mu$CBS (p<0.00001, rank-sum test).

## 4   Discussion

We have examined several selection methods for efficiently recording from a multiplexed electrode array. Existing heuristic methods like the Checker pattern, although offering better spatial coverage

than sampling densely, do poorly in recovering the gamut of neurons accessible via the electrode array. The CBS routing method, in contrast, yields approximately the same number of neurons that would have resulted if every electrode had been recorded simultaneously, yielding 41% more neurons than the best-performing heuristic. CBS and $\mu$CBS also used electrodes more efficiently than other selection methods, favoring electrodes with higher SNR.

In CBS, recording electrodes are selected based purely on the neurons' waveforms, and not on the information content of their spike trains. Although this strategy is unbiased and desirable in some settings, we present another method that prioritizes neurons with utility in decoding variables of interest, such as sensory stimulus properties, cognitive states, and movement kinematics. A modified algorithm can be found in the supplementary materials.

The random greedy swap approach used in CBS to optimize the selection map was used primarily for its efficiency. Future work might use other combinatorial search methods such as simulated annealing that trade off speed and quality of the selection map.

This work is related to channel selection methods developed for EEG [9, 19] and ECoG [26] applications. These existing methods select channels to reduce dimensionality as a preprocessing step for decoding or to identify physiologically active channels. CBS, in contrast, does not attempt to reduce the number of recording channels, but instead chooses how to route a fixed number of channels to available electrode sites given wiring constraints. CBS also chooses selections to maximize waveform separability in single unit recordings, whereas the existing methods operate on field potentials. Another related area of work deals with high-dimensional feature selection using a separability criterion. Lei et al. [10] showed that features could be selected using an approximate pairwise LDA criterion. In CBS, we avoided using a simplified objective and instead increased efficiency by reducing waveform dimensionality using PCA and exploiting bandedness when inverting $S_w^\theta$. These measures are appropriate since spike waveforms are well approximated by a small number of components, and spikes are physically limited in their spatial extent.

In this work, we use Kilosort2 for spike sorting because it represents the current state-of-the-art [12]. All spike sorting algorithms approximate the ground truth set of neurons, and our results should be interpreted with the capabilities of Kilosort2 in mind. We have mitigated concerns solely due to spike sorting algorithm performance by determining neuron yields following a step of human curation. Another concern is that serial validation may overestimate neuron counts compared to true simultaneous recordings due to edge effects at bank boundaries. This concern applies equally to all selection methods and only affects interpretation of the *absolute* values for neuron counts. We note, however, that only 2.0% of neurons contacted a bank boundary (had boundary-adjacent channels with amplitude > 20% of max). We mitigated other concerns about the influence of probe drift on our results by stabilizing the brain surface, waiting 30-60 minutes after probe insertion before recording, and studying relatively short duration recordings (average 6.3 minutes).

CBS can straight-forwardly be incorporated into existing recording workflows with relatively minimal time burden. For example, in our Neuropixel experiments, the training recordings took 6 minutes, spike sorting took $\sim 5$ minutes, and the CBS-based optimization took $\sim 3$ minutes on a consumer PC, for a total typical run time of 14 minutes.

We have discussed CBS in terms of finding a *single* best selection map. CBS can also generalize to finding *multiple* non-overlapping selection maps. These could be particularly useful in a time division multiplexing (TDM) scheme [24, 17, 2], in which a sequence of maps $\theta_1, ..., \theta_M$ is repeated every TDM cycle. CBS potentially allows $M$ to be far less than the naïve number of electrodes needing to be sampled, thus increasing the available sampling rate per channel. Another area that merits further work involves periodically updating selection maps to account for probe drift or the emergence of new neurons.

## Acknowledgments and Disclosure of Funding

This work was supported in part by an award from the Simons Foundation (SCGB Award 325548), the National Institutes for Health (R01-NS104923, U01-NS0990577, U01-NS103518), the Army Research Office (68984-CS-MUR) and the Defense Advanced Research Projects Agency (N66001-17-C-4002).

## Broader Impact

This report describes a technique that can improve the effectiveness of brain-machine interfaces. As devices such as electrode arrays scale up in their capabilities and their safety, new opportunities emerge for treating neuropsychiatric conditions. Treatments that monitor the activity of thousands or millions of neurons to deliver precise feedback control of aberrant activity could address a host of brain disorders when treatment is pharmacologically intractable. This strategy is already being applied in experimental treatments for movement disorders and the early detection and prevention of seizures, potentially improving the quality of life for tens of millions of patients globally. We envision that eventually, mood disorders or conditions where inter-areal communication channels in the brain are lost or dysfunctional could one day be repaired using large-scale neural recording and stimulation techniques. However, new devices that record massive amounts of neural data could expose the user to breaches of privacy in a way that has not been encountered previously. Rich brain activity data, unfiltered by intention or the bounds of normal communication, could be accessed covertly by malicious actors or state surveillance. Developing a security model should be a high priority for these devices as they continue to translate into the clinic, especially as these devices will likely be connected in some way to the internet.

More broadly, optimal channel selection could benefit other areas of neuroscientific research. Our method leads to the recording of 41-85% more neurons per unit time over previous methods when using the Neuropixel probe. The Neuropixel is being widely adopted in labs across the field, and it is being used as a standard tool in large collaborative efforts such as the International Brain Lab [8]. Other neural-matrix style probes for neural spike-band recordings [17] as well as high-density electrocorticographic (ECoG) arrays can benefit from electrode selection [1]. In all cases, our method can shorten the amount of recording time needed for producing the same scientific conclusions. Reduction of required experimental time, or reduction in the number of animals needed for a study, would also be welcome from an animal welfare perspective.

Our algorithm confers the most benefit to specific neurotechnologies that face a physical bottleneck, which in the case of the Neuropixel, is wiring. Many of these new devices are an order of magnitude more costly than previous low-bandwidth counterparts. As with all new neurotechnologies, our method might most immediately benefit labs or individuals with more financial resources. In the long-term, however, we hope that these new technologies will become more widely accessible.

## Footnotes

[1] https://github.com/pesaranlab/neuro_cbs

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
