[Supplementary Material]



Figure S1: A: TaskScore frequency histogram. B: Mean change in recorded spike energy relative to unmodified CBS. Neurons are grouped via `TaskScore` quantiles. Error bars signify $\pm 1$ SEM.

## Prioritizing information-rich neurons

### Modified algorithm

In CBS, recording electrodes are selected based purely on the neurons' waveforms, and not on the information content of their spike trains. Our algorithm can be straight-forwardly extended to bias electrode selection towards neurons with greater relevance to a behavioral task or stimulus. We first quantify this relevance with a metric denoted as `TaskScore` $\geq 0$. For the $i$th neuron, `TaskScore`$(i)$ reweights the neuron's contribution to the scatter matrices used in CBS as

$$ S_w^\theta \to \sum_{i=1}^{N} \widetilde{\alpha}_i \Sigma_i^\theta \text{ and } S_b^\theta \to \sum_{i=1}^{N} \widetilde{\alpha}_i (\mu_i^\theta - \mu^\theta)(\mu_i^\theta - \mu^\theta)^T, $$

where $\widetilde{\alpha}_i = \alpha_i / \sum_j \alpha_j$ and $\alpha_i = $ `TaskScore`$(i) + \beta$. $\beta \in [0, \infty]$ is a smoothing parameter that controls how much task-relevance is weighted. If $\beta = 0$, task information dominates selection and the weights are the task scores. When $\beta \to \infty$, we recover the original task-blind score and each neuron is weighted equally.

### Experimental validation

We validated this modified selection algorithm on recordings made while the animal performed center-out reaching movements to 7 peripheral targets on a touch screen. For each reach direction, spikes for each neuron were counted in a window from 200 ms prior to and 800 ms following target presentation and averaged across trials. We denote this average spike count for neuron $i$ and reach direction $j$ as $\tau_j(i)$, and set

$$ \texttt{TaskScore}(i) := \frac{\max\limits_{j} \tau_j(i) - \min\limits_{j} \tau_j(i)}{\max\limits_{j} \tau_j(i) + \min\limits_{j} \tau_j(i)}. $$

`TaskScore`$(i)$ is set to zero if the denominator above is zero. Figure S1A shows the distribution of `TaskScore` values for the neurons studied.

We quantified the modified algorithm's effect on how densely each neuron was monitored by using a normalized spike energy measure. This measure is the sum of the peak-to-peak spike amplitudes for all selected electrodes, divided by the sum of all spike amplitudes on all electrodes. As a neuron is more densely monitored, i.e., as more electrodes surrounding a neuron are selected, the neuron's normalized spike energy approaches 1. We computed the normalized energy for all neurons under different values of $\beta$ (Fig. S1B). When $\beta = 0$, the top 25% most task-relevant neurons were more heavily monitored, resulting in a 1.1% higher recorded spike energy (p = 0.001, one-sample t-test),

whereas task-irrelevant neurons (bottom 25%) were less densely monitored, incurring a 2% decrease in spike energy ($p = 0.018$, one-sample t-test). Interestingly, while statistically significant, the effect of the bias towards task-relevant neurons was relatively small in our data, suggesting that task-blind CBS sufficiently monitors almost all available neurons.