[Reviews · NeurIPS 2020]

Review 1

Summary and Contributions: The manuscript is addressing a challenge in the high-density electrode field where each recording channel can be routed to one of several electrode sites on an array. Multiplexing is a strategy employed for very high density arrays but a question remains on how to distribute limited channels with plentiful electrodes at any given time. The authors state that the current technologic state is such that no adaptive methods exist to use recorded neural data to optimize/customize the electrode selections per recording context. They present an algorithm called classification-based selection (CBS) that optimizes the joint electrode selections for all recording channels so as to maximize isolation quality of detected neural signals. In experiments using Neuropixels in non-human primates, they show their algorithm yields a similar number of isolated neurons as would be obtained if all electrodes were recorded simultaneously. "Neuron counts were 41-85% improved over previously published electrode selection strategies." The neurons isolated from electrodes selected by CBS were a 73% match, by spike timing, to the complete set of recordable neurons around the probe. The electrodes selected by CBS exhibited higher average per-recording-channel signal-to-noise ratio. The authors state that CBS, and selection optimization in general, could play an important role in development of neurotechnologies for BMI, as signal bandwidth becomes an increasingly limiting factor.

Strengths: The strengths are quite clear. There does exist a bandwidth problem in BMI and particularly high density arrays. Each year there are improvements in the core technology but mastery of the data captured in such situations presents many elusive problems. BMI is a field which will only continue to grow and the clinical applications at this point are open to virtually all neurologic disease states.

Weaknesses: I would add that electrode selection algorithms do indeed exist although perhaps in different contexts. One example is Saboo et al 2019 (Saboo, K.V., et al. Unsupervised machine-learning classification of electrophysiologically active electrodes during human cognitive task performance. Sci Rep 9, 17390 (2019)). There are many others and of course iEEG is not the same as single unit recording although principles of channel selection still apply. Simply put, the authors over-state the novelty of the algorithm somewhat.

Correctness: The claims are largely correct aside from that mentioned above. The methodology is sound.

Clarity: The paper is well written, has logical flow and consistently follows from one aspect of proof to the next, no major issues.

Relation to Prior Work: This aspect of the manuscript could be better and this is addressed above. Statements such as "Our algorithm confers the most benefit to new neurotechnologies that face a data bottleneck." line 335 are very strong. There are methods beyond Checker and Kilosort2 (see david redish spike sorting methods and packages).

Reproducibility: Yes

Additional Feedback:


Review 2

Summary and Contributions: This paper introduces an algorithm to select an informative subset of channels from a multi-contact neural probe (on many of which it is not possible to record all channels simultaneously), so as to increase the yield of identifiable neurons. The new approach is based on greedy optimization of a heuristic cost function (Fisher's linear discriminant) based on activity recovered from a pilot "training" period of dense recording, and so contrasts with many current approaches which use a fixed electrode selection pattern.

Strengths: The proposed approach is a sensible and easy-to-implement solution to a practical problem in experimental neuroscience. Experimental validation suggests that it might work to yield recordings of similar quality to that that would be possible if all channels could be recorded (although the metric used in the experiments is odd -- see below).

Weaknesses: My main concern is that the algorithm presented seems to be most relevant to neuroscience experimentalists, rather than to the theoreticians, data modellers and machine learners who are more commonly numbered in the NeurIPS community. While some specific choices are dictated by the hardware at hand, theoretically the method is straightforward and far from novel. The cost function, a form of Fisher's Linear Discriminant, dates to 1936. Others have based feature selection methods on the discriminant cost (e.g. Z. Lei, S. Liao and S. Z. Li, "Efficient feature selection for linear discriminant analysis and its application to face recognition," Proceedings of the 21st International Conference on Pattern Recognition (ICPR2012), Tsukuba, 2012, pp. 1136-1139). Thus, it is difficult to see this being of strong interest at NeurIPS. I did not understand the focus on number of neurons discovered in the experiments. The workflow, as I understand it, requires that neurons be identified using the dense recordings first. Thus, rather than re-running spike sorting on the reduced channel data, all that is needed is to detect waveforms and assign them to the recognized neurons (perhaps adding adaptive methods to track drift and isolate neurons that first spike later in the recording). The metrics should focus on this task.

Correctness: I did not spot any errors.

Clarity: Being unfamiliar with the details of multichannel probes, I found parts of the paper difficult to follow. The geometry of the banks is not clear; nor was it obvious until I reverse engineered the counting calculation on page 2 what it meant to "assign an electrode to a bank". It may have helped to have shown the banks and potential assignments of a single channel in Figure 1. The fact that densely recorded training data were required was not brought up until quite late, leaving me puzzling over how the within- and between-class scatters could be constructed. It would have been helpful to have laid out all the stages of the approach in a figure or algorithm box at the outset.

Relation to Prior Work: The link to LDA/Fisher and previous work on feature selection in general is not discussed explicitly.

Reproducibility: Yes

Additional Feedback: In response to author feedback: I remain unconvinced that this work is of sufficient significance for NeurIPS -- the authors' response does indeed make clear that they have not exactly reproduced the approach of Lei et al, but the technical advance remains incremental. Their response re: the # neurons metric misses the point, I think. The number of neurons discovered is defined by the "dense" setting (where the initial spike-sorting is performed) -- that is not a feature of the subsampling scheme (one could use the same approach for any selection scheme); the point of subsampling is to keep the waveforms from those neurons distinct.


Review 3

Summary and Contributions: This paper presents a method for choosing which channels of a dense microelectrode array to choose to maximize the number of units recorded, and presents a validation on Neuropixel data.

Strengths: The biggest strength of this work is relevance -- overcoming hardware constraints with smart machine learning is a great idea and very relevant to the age of big data in neuroscience. Furthermore, the approach is theoretically simple and easy to understand, with clear gains.

Weaknesses: The biggest weakness I can see of this work is that NeurIPS may be the wrong conference. The people who need to see this are the experimentalists first, and then the ML community to see if they can improve on the procedure once a good benchmarking dataset has been proposed. The theoretical idea is equivalent to solving the Fisher's discriminant analysis cost function over the set of binary projections of a certain dimension, which is not dramatically new, and so the really exciting thing about this work is a novel field of application. Next, the discussion does mention how important Kilosort2 is to the results they show, which is good (though I think it should be mentioned earlier), but it does not mention the importance of behavior. Finding a good subset of channels for neurons during freeform rest may not translate well into neurons active during a task, and this relationship is important to the use of an algorithm like this in practice.

Correctness: The methodology is clearly described and seems reasonable to me.

Clarity: I find the paper easy to understand and intuitively structured.

Relation to Prior Work: The paper makes the claim that it is the first in this field.

Reproducibility: Yes

Additional Feedback: As a neuroscience methods paper this is a strong accept from me, and my comments are mostly related to small things. My concerns are related to audience, since this requires significant neuroscientific background to understand, hence why I only give it a "good" here. ------ My score has not changed in light of the author feedback. Specific questions: -- The notation is a bit confusing around the x_{\theta}. It is mentioned later in the paper that there are 100 random samples of each waveform used to compute the within and between scatter matrices described in the theory section, but that is not well-described in the first place. This implicit index is somewhat confusing. Similarly, readers not well-versed in neuroscience may find it confusing that a large implicit step in the theory is a black-box spike sorter that is applied between the electrode and channel (I think?) level -- being more explicit about the relationship there would be helpful. -- I'm confused by the use of PCA in line 147 -- this would mean that each channel has different PCA components, wouldn't it be preferrable to take all waveforms across all channels and do the temporal PCA with that as input? -- line 179: I find this statement puzzling. So you greedily choose the N_c electrodes that have the highest AmpScore? I don't understand how the banks fit into this problem since you seem to be implying that they are big contiguous binary vectors you premultiply your v(t) by, but I didn't see an explicit definition of them. -- line 219: What are the parameters of the classification task here?


Review 4

Summary and Contributions: The paper describes an approach for electrode selection for a channel limited microelectronics recordings with the goal of maximizing the number of single neurons recorded from. The approach casts this combinatorial optimization as a feature selection problem for multi-class classification using the objective for Fisher linear discriminant analysis. This objective is motivated by the case that the different electrodes provide additional feature dimensions for discerning among the neural waveforms, and the class conditional distribution is a multivariate Gaussian with the average waveform (template) as the mean. The paper proposes a greedy pairwise swap based combinatorial search, and also uses approximation that ignores the covariance of electrodes to define a fast approximation that maximizes a signal-to-noise ratio. Empirical evaluations are used to compare versus benchmarks corresponding to regularly patterned electrode selection. The results show increases in number of neurons recovered, better precision in the spike train precision. A gold-standard of manually curated neurons was also assessed. Overall, the paper describes how a standard approach from supervised learning can be applied to address a specific problem in neural engineering.

Strengths: The approach is sound and has the possibility of becoming the current state of the art in adaptive electrode selection. The comparison of the simpler approach and standard electrode layouts provide reasonable contrast for the empirical evaluation.

Weaknesses: My main concern is that the empirical results consist of 10 sessions for the same subject/animal in premotor cortex. These 10 sessions define the sample size of the statistical tests. Is the sample of sessions representative of general performance? A key weakness that is discussed (last paragraph of discussion) is that the objective is not tied to the information content of the spike trains.

Correctness: Empirical evaluation is correct but limited.

Clarity: Yes, the paper is clearly written.

Relation to Prior Work: Yes.

Reproducibility: Yes

Additional Feedback: In section 3, it is not clear from context what is meant by "median-subtracted separately from each channel". Does that mean the template is the median of the occurrences, or that the amplitude of the template is taken as the median. In terms of approach, it seems that simulated annealing would be an alternative to the greedy swap step in the random plus greedy swap approach. In the discussion, it is not clear how time division multiplexing would be incorporated. Perhaps some more information on the rate at which this multiplexing would occur would be important if this is referenced as future work. In the broader impact, it is not clear that the recording leads to "more meaningful brain activity", as that would require assessing task relevance. ++++++++++++++++++++ The authors have provided evidence in the rebuttal that the objective can incorporate 'information' regarding task relevance by reweighting the scatter matrices. This pertains to a weakness I mention. I will maintain my current score.

[Author Response · NeurIPS 2020]

We thank the reviewers for their feedback and suggestions. We appreciate this opportunity to clarify important aspects of our work and to describe proposed improvements to our manuscript that address the reviewers' concerns.

**1. Intellectual fit at NeurIPS:** We are very excited for the opportunity to present our work introducing the new application area of *adaptive electrode selection algorithms* to the NeurIPS community. NeurIPS has already been instrumental in advancing algorithms for applications in neuroscience and "5. Neuroscience and Cognitive Science" (subcategories: Brain Imaging and Brain–Computer Interfaces), and "2. Applications" are main categories in the call-for-papers. NeurIPS should now promote the area of adaptive electrode selection algorithms that comprise a growth area in neurotechnology. High-quality datasets are essential for developing new applications (e.g., ImageNet, MNIST, etc.). In this work, we collected and will publicly release a first-in-class, curated dataset. Therefore, along with introducing a new application area, our work provides a benchmark data set to support further development. Finally, we present CBS, which we show is a highly-effective, fast, reliable algorithm that successfully monitors almost all the available neurons. Thus, we propose our work has excellent intellectual fit with NeurIPS.

**2. Incorporating behaviorally-informative neurons:** Multiple reviewers raise a very important point: electrode selection can also target behaviorally-informative neurons. We have now extended our work to prioritize neurons with greater task relevance, quantified by a metric denoted as `TaskScore`. `TaskScore` reweights contributions of neurons to the CBS objective: $S_w^\theta = \sum_{i=1}^N \widetilde{\alpha}_i \Sigma_i^\theta$ and $S_b^\theta = \sum_{i=1}^N \widetilde{\alpha}_i (\mu_i^\theta - \mu^\theta)(\mu_i^\theta - \mu^\theta)^T$, where $\widetilde{\alpha}_i = \alpha_i / \sum_j \alpha_j$ and $\alpha_i = \texttt{TaskScore}(i) + \beta$. $\beta \in [0, \infty]$ is a smoothing parameter that controls how much task-relevance is weighted. If $\beta = 0$, task information dominates selection and the weights are the task scores. When $\beta \to \infty$, we recover the original task-blind score and each neuron is weighted equally. We have implemented this extension. Figure 1 presents the results. When $\beta = 0$, the top 25% most task-relevant neurons were more heavily monitored, resulting in a 1.1% higher recorded spike energy, whereas task-irrelevant neurons (bottom 25%) were less densely monitored, incurring a 2% decrease in spike energy. Interestingly, while statistically significant, the effect of the bias towards task-relevant neurons was relatively small in our data, suggesting that task-blind CBS sufficiently monitors almost all available neurons. As intuited by the reviewers, our results suggest that task-relevant weighting will have larger impact when a smaller fraction of the available neurons can be recorded. We will include this extension as a supplementary section.

Figure 1: Mean change in recorded spike energy relative to unmodified CBS. Neurons are grouped via `TaskScore` quantiles.

**3. Significance and relation to earlier work:** We agree with Reviewers #2 and #3 that Fisher's Linear Discriminant (FLD) has a long history and the connection to FLD should be explicit. We will amend the text appropriately. Nevertheless, as recognized by the reviewers, new applications, solvers, and approximations based on FLD can provide meaningful and important extensions to earlier work. Reviewer #2 points to the relevant work of Lei et al. 2012, who present a greedy approach to FLD-based feature selection. We will add this citation and discussion. We note that Lei et al. improve efficiency via a simplified criterion, while we use the full criterion and instead exploit 1) PCA to reduce waveform dimensionality and 2) banded structure for inverting $S_w^\theta$. The differences between application areas should also be noted. In the Lei et al. study, the application (face recognition) leads to use of a simplified criterion, whereas this is not required for our application.

Reviewer #1 also brings up work on ECoG electrode selection (Saboo et al. 2019) that selects task-relevant electrodes without constraints, like in region-of-interest (ROI) estimation. We will add this citation. We note that we address a physical bottleneck that arises in Neuropixels and other high density arrays. Unlike in Saboo et al., we seek the jointly optimal "view" of the neuronal population under physical constraints. This differs from sub-selecting all highly relevant electrodes in the absence of constraints (i.e., wiring or bandwidth).

**4. Success metric & use of training data:** Reviewer #2 questions the focus on the # of neurons discovered and recommends matching sub-sampled waveform data to training templates as an alternative to metrics computed after re-running spike-sorting. We should first clarify that we focused on maximizing the # of discovered neurons because this is a fundamental goal which supports all subsequent population-level analyses. We further clarify that in one of our metrics, we find matches in spike timing between sub-sampled and densely recorded neurons. We also see the value of the suggested metric since it directly evaluates sub-sampled waveforms with their training templates. We thank the reviewer for their insight and will add this as part of the validation.

**5. Bottleneck clarification:** Reviewer #1 was concerned about the data bottleneck of spike-sorting and recommended fast spike-sorting methods. We appreciate the opportunity to clarify that the bottleneck we reference in that statement is not a computational bottleneck in spike-sorting, but a data-acquisition bottleneck in modern devices. We will clarify this point in the revision.

**6. Clarifications:** We appreciate all the reviewers' recommended points to clarify (i.e., notation, probe layout, median subtraction, alternative ascent strategies, etc.) We will update and improve the manuscript to address these points.

[Meta-Review · NeurIPS 2020]

This paper investigates a new problem for neural information processing in the age of large-scale recording. The proposed algorithm is relatively simple (which is a positive). With promising application and openly available code/data, summing up reviewers opinions, this paper could be accepted.